# A Food-Safety Risk Assessment of Mercury, Lead and Cadmium in Fish Recreationally Caught from Three Lakes in Poland

**DOI:** 10.3390/ani11123507

**Published:** 2021-12-09

**Authors:** Agnieszka Chałabis-Mazurek, Jacek Rechulicz, Renata Pyz-Łukasik

**Affiliations:** 1Department of Pharmacology, Toxicology and Environmental Protection, University of Life Sciences in Lublin, 13 Akademicka Str., 20-950 Lublin, Poland; agnieszka.mazurek@up.lublin.pl; 2Department of Hydrobiology and Protection of Ecosystems, University of Life Sciences in Lublin, 37 Dobrzańskiego Str., 20-262 Lublin, Poland; 3Department of Food Hygiene of Animal Origin, University of Life Sciences in Lublin, 13 Akademicka Str., 20-950 Lublin, Poland; renata.pyz@up.lublin.pl

**Keywords:** heavy metal, water, sediments, fish meat, food safety

## Abstract

**Simple Summary:**

The research aimed to determine the content of lead (Pb), cadmium (Cd), and mercury (Hg) in water, sediment, and freshwater fish species roach (*Rutilus rutilus*), perch (*Perca fluviatilis*), and pike (*Esox lucius*) from the Dratów, Syczyńskie, and Czarne Sosnowickie lakes located on Polesie Lubelskie, Poland, as well as a food safety assessment for the consumer. Pb and Cd were measured by graphite furnace atomic absorption spectrometry, while Hg was measured by cold vapor atomic absorption spectrometry. The research results showed that both the waters and the sediments of the studied lakes are characterised by a low concentration of Pb, Cd, and Hg, which indicates the lack of a moderate influence of anthropopressure on these reservoirs. The range of heavy metal contents in the muscles of roach, pike, and perch for Pb was 0.0399–0.1595, 0.0305–0.0920, and 0.0296–0.1057 mg kg^−1^, respectively; for Cd 0.0014–0.0095, 0.0010–0.0015, and 0.0020 mg kg^−1^, respectively; and for Hg 0.0123–0.0499, 0.0185–0.0255, and 0.0216–0.0583 mg kg^−1^, respectively. The content of heavy metals in fish muscles was low and conformed to requirements as defined in the European Union (EU) food legislation. The health risk assessment with regard to the heavy metal contents in the muscles of fish confirmed the safety of this food for consumers.

**Abstract:**

Heavy metals are introduced into water due to anthropogenic activities and can significantly affect an entire ecosystem. Due to their close integration with the water environment, fish are a sensitive indicator of contamination. In addition, fish is an important element in human diets, therefore, monitoring the concentrations of metallic contaminants in their meat is particularly important for food safety. This study aimed to assess the pollution of water ecosystems with selected toxic heavy metals in lakes Dratów, Czarne Sosnowickie, and Syczyńskie. The concentration of Pb, Cd, and Hg in water, sediment, and freshwater fish muscle tissue was determined, and a food safety assessment was performed. The analysis of water and sediments showed that the sediments were characterised by a significantly higher concentration of heavy metals. Presumably, this ecosystem element plays an important role in the uptake of heavy metal contaminants by fish whose levels were higher in planktonophagous and benthophagous fish species as compared to predatory fish. The food safety assessment showed that amounts of heavy metals in the muscle tissue posed no threat to the health of consumers ingesting that fish species, neither individually (THQ) nor collectively (TTHQ).

## 1. Introduction

Fish are important organisms of the aquatic ecosystem because they are one of the largest animals inhabiting this environment while at the same time, occupying different levels of the trophic pyramid [1,2]. In addition, fish are crucial for the management of water resources (fishery management) and recreational use [3]. Amateur-caught fishes are valuable fishing trophies and are very often used as cuisine, being part of the human diet and a valuable source of protein [4,5]. This phenomenon applies especially to locally occurring reservoirs, where local peoples often use the small reservoirs to obtain fish.

Due to the diversity of nutritional requirements, food selectivity, and food availability of various fish species, their nutrition is closely related to their living environment [6,7]. The quality of water and bottom sediments as abiotic elements shaping the conditions in the aquatic ecosystem is of particular importance [8,9,10,11].

Heavy metals are one of the most significant civilisation pollutants. Their negative significance is widely known and many of these metals should be constantly monitored in the environment [12,13]. The human consumption of water resources contaminated by heavy metals may have profound health implications. For example, Cd is persistent and can be stored in fish muscles, causing seafood poisoning in humans. It may cause health problems related to the weaker functioning of the body’s internal systems and the emergence of diseases in the form of cancer of the bone, heart, kidney, etc. [14]. Accumulation of Pb in organisms weakens the functioning of the immune system, circulation, and reduces hormonal and enzymatic activity [15]. Therefore, heavy metals such as Hg, Cd, and Pb, among others, are classified as toxic, and maximum permissible levels are set for human consumption [16,17].

Fish are often the last level of the trophic pyramid in aquatic ecosystems and therefore, are good bioindicators for heavy metals [18,19,20]. Heavy metals get into fish organisms by staying in the aquatic environment and penetrating vegetation and bottom sediments where the fish search for food (omnivorous and benthivorous fish) or by predatory fish eating these other fish. The presence in the water environment and their food is of decisive importance in the possibility of heavy metals entering and accumulating in the fish body and the muscles that are food for humans [3,21,22,23].

Besides their ecological significance, fish in aquatic ecosystems are important for hobby and recreation activities. In recent years amateur fishing has developed a great deal and has become a significant way to spend time outdoors. In addition, fishing trophy fish are often consumed by anglers and their families [5]. Fish caught as trophies that humans consume can become a source of heavy metals and, through their accumulation, can pose a threat to people’s health. As a result, the safety of food from this source may be questionable [7,24,25,26,27,28,29,30].

Therefore, the purposes of the present study were to determine: (I) does the living environment (i.e., different lakes and the concentration of heavy metals in water and sediment) affect the content of heavy metals such as Pb, Hg, and Cd in fish muscles; (II) whether the species of fish and their trophic level affect the heavy metal content in the meat of fish; and (III) whether the concentration of these heavy metals in the muscles poses a health risk when fish, caught by amateur fishing, may be a potential food source for humans.

## 2. Materials and Methods

### 2.1. Study Area

The fish for the analysis of heavy metals such as Pb, Cd, and Hg came from three lakes localised in the Polesie Lubelskie region of eastern Poland: Dratów, Syczyńskie, and Czarne Sosnowickie. Two of these lakes, Dratów and Czarne Sosnowickie, are included in the Wieprz-Krzna Canal (W-KC) system and over 70–80% of their catchments are used for agriculture purposes (mostly arable lands and meadows). The third lake, Lake Syczyńskie, is very small, located in the centre of the village of Syczyn, and its relative area has a large catchment area (about 460 ha) and over 86% of it is used for agriculture (arable fields, meadows, and pastures). According to the classification of Rücker et al. [31], Lake Syczyńskie is a “Planktothrix-lake”. As shown by Wiśniewska et al. [32], in summer and autumn, *Planktothrix. agardhii* comprises approximately 96% of the potentially toxic Cyanobacteria in this lake. The detailed data on the characteristics of the lakes and their water quality is presented in Table 1. The selected lakes are distant from each other (Figure 1), they have different fishing pressure, and are characterized by different ecological statuses.

The users of lakes Dratów and Czarne Sosnowickie are the Polish Angling Association Lublin District (PAA Lublin) and Lake Syczyńskie—PAA Chełm District. In all the studied lakes, fishery management is conducted based on fish restocking and the fishery use is limited to only amateur fishing (Table 1).

### 2.2. Collection of Water, Sediment, and Fish Samples

Triplicate superficial water samples were collected at a half meter depth from the water surface from the coastal zone in 10 designated sampling sites for each lake. Water samples were collected in plastic vessels and filtered using a 0.45 µm Millipore filter. In the same sampling places, triplicate samples of bottom sediments submerged at least 30 cm below the water level were collected using a plastic tubular spoon, air-dried, ground, sieved using a 1 mm nylon sieve, and then stored in clear plastic containers for future analysis. The data on general information from 2014–2018 related to the volume of angling catches and the number of anglers were obtained from lake users (PAA Lublin and PAA Chełm). Based on these data, the size of fish catches in individual reservoirs was estimated. The overall abundance and biomass of caught fish per year and the share of predatory and non-predatory species in the abundance and biomass structure were determined. Moreover, several relative parameters related to the angling use of these lakes were calculated, such as the average number and biomass of fish caught per 1 ha of the reservoir per year, the average annual number of anglers per 1 ha of the reservoir, and the number and biomass of fish caught by one angler during the year.

The fish for analysis of Pb, Cd, and Hg concentration contents in muscles were caught in 2018 (July–September) using standard monitoring methods using sets of Nordic nets. Out of all the caught fish, the three most popular species for amateur fishing were selected: roach (*Rutilus rutilus*), perch (*Perca fluviatilis*), and pike (*Esox lucius*). All selected fish for analysis were measured (within an accuracy of 1 mm) and weighed (within an accuracy of 1 g). The muscle tissues were collected from each fish for analysis for the content of heavy metals Pb, Cd, and Hg. For measurements of heavy metals, the muscle tissues were collected, homogenised, put into polyethene bags, and kept frozen at −20 °C until analysed.

### 2.3. Metal Analysis

The determination of total Hg in superficial water, bottom sediments, and muscular tissues were measured by cold vapour atomic absorption spectrometry without pre-treating the samples. For this purpose, water, sediments, and homogenised muscles were directly weighed (10–100 ± 0.1 mg) into pre-cleaned combustion boats and inserted into the automatic Mercury/MA-2000 analyser (Nippon Instruments Corporation, Tokyo, Japan) [33]. The method was controlled by analysing the series of samples from a certified reference material (BCR-463 tuna fish, IRMM, Geel, Belgium). The following validation parameters were determined: detection limit, quantification limit, recovery, trueness, and repeatability/precision which were 0.096 ng; 0.192 ng; 96%; 4.0%, and 1.2%, respectively.

Before Cd and Pb determination, 1 g sediment samples and 5 g of homogenised fish muscles (weighed within an accuracy of ±0.0001 g) were digested using a microwave stove Multiwave 2000 (Anton Paar, Graz, Austria). The levels of Cd and Pb were assayed by using an atomic absorption spectrometer (GFAAS) with electrothermal atomisation and Zeeman background correction (SpektrAA 220Z, Varian, Mulgrave, Victoria, Australia) [34]. A palladium solution (Merck, Darmstadt, Germany) was used as a chemical modifier to analyse the Cd and NH_4_H_2_PO_4_ (Merck, Darmstadt, Germany) for Pb. In order to determine the validity of the method, DORM-3 (Fish Protein Certified Reference Material for Trace Metals, NRCC, Ottawa, Canada) was subjected to the same analytical procedure and was tested for accuracy.

The method’s detection limit, quantification limit, recovery, trueness, and repeatability/precision were 0.001 µg/g, 0.002 µg/g, 91.6%, 8.4%, and 0.56%, respectively for Cd and 0.001 µg/g, 0.002 µg/g, 109%, 9%, and 1.2%, respectively for Pb.

### 2.4. Health Risk Assessment

The human health risk from fish consumption was estimated for each element using the following equations [35]:EDI = Cm × dcfish/bw
THQ = EDI/RfD
TTHQ = THQ (Hg) + THQ (Pb) + THQ (Cd)
where:

EDI—estimated daily intake (µg kg^−^^1^ per day),

Cm—mean concentration of elements in the fish muscle (µg g^−1^ = mg kg^−^^1^),

dcfish—the daily per capita consumption of freshwater fish in Poland (5.47 g per capita per day; www.fao.org/faostat/en/#data/CL) (accessed on 5 July 2021),

bw—average adult human body weight (70 kg),

THQ—target hazard quotient,

TTHQ—total target hazard index,

RfD—established reference dose (µg kg^−^^1^ per day) [35].

### 2.5. Statistical Analysis

To compare the angling use of the studied lakes, an analysis of relative indicators was performed. For the studied lakes, the number and weight of caught fish per 1 ha of the reservoir per year, the number of anglers per 1 ha of the reservoir per year, and the number and weight of fish per 1 angler in a year were compared using the analysis of variance (one-way ANOVA, factor: lake). The concentration of Pb, Cd, and Hg in water, sediments, fish muscles, and fish food guilds (predatory and non-predatory fish) was compared within each of the studied lakes using a one-way ANOVA, factor: lake. In addition, the lakes were compared with each other using the two-factor analysis of variance (ANOVA, factor: lake; water or sediment or fish guilds or fish species). The Spearman correlations between the content of the tested heavy metals in sediments and water and their content in the muscles of the fish were calculated. All statistical analyses were performed using the Statsoft Statistica 13.3 package with a significance level at *p* ≤ 0.05.

## 3. Results

### 3.1. Fish and Angling Use of Studied Lakes

Among the tested fish, the pike was characterised with the highest average total length, while the average length of the two other fish species ranged from 13.5 to 23.1 cm. Detailed data on the size structure of the analysed fish species are presented in Table 2. The studied species of fish had a different share in amateur fishing catches. An analysis of the results showed that the studied fish species were most present in the results for amateur fishing in Lake Syczyńskie (over 79% in abundance and 73% of the biomass for all caught fish). The lowest share of *R. rutilus*, *P. fluviatilis,* and *E. lucius* in the results for amateur fisheries was observed in Lake Dratów (Table 2).

An analysis of the angling use of lakes showed that a much greater number of fish per 1 ha of the lake was caught in Lake Syczyńskie (*p* < 0.05). Additionally, the largest fish biomass (25.93 kg year^−1^) was caught from this lake as well, but compared to the other two lakes, this difference turned out to be statistically insignificant (*p* > 0.05) (Table 3). An analysis of the number of anglers using these lakes for amateur fishing showed that the largest number of anglers per 1 ha of the lake during the year were also recorded in Lake Syczynskie, i.e., on average 2.63 persons year^−1^ (*p* < 0.05). In total, from the three studied lakes, anglers caught and took home on average 12 individuals per year in Lake Czarne Sosnowickie to over 19 individuals in Lake Dratów. The biomass of fish taken by anglers ranged from nearly 5 kg of angler^−1^ year^−1^ for Lake Czarne Sosnowickie to almost 10 kg of angler^−1^ year^−1^ for Lake Syczyńskie (*p* > 0.05). The detailed data on the angling use of the lakes are presented in Table 3.

### 3.2. Heavy Metals in Water and Sediments

An analysis of the concentration of heavy metals (Pb, Cd, and Hg) in water and sediments within the lakes showed that in almost every lake, significantly higher concentrations were recorded in the sediments (*p* < 0.05) (Table 4). Despite the large differences in the content of the examined heavy metal elements, the analysis did not prove statistical differences in the Pb content in the sediments and water in Lake Dratów, as well as the accumulation of Hg in water and sediments in Lake Syczyńskie. The determined concentration of heavy metals in the sediments showed that by far the highest values of this parameter (*p* < 0.05) were recorded for the sediments in Lake Czarne Sosnowickie, on average for Pb = 45.823 mg kg^−1^, for Cd = 0.477 mg kg^−1^, and Hg = 0.021 mg kg^−1^ (Table 4). Moreover, the lakes differed in the Hg content of the water and the highest value was recorded for Lake Czarne Sosnowickie (average 0.0050 mg dm^−3^) and the lowest value for the water from Lake Dratów (average 0.0016 mg dm^−3^) (*p* < 0.05) (Table 4).

### 3.3. Heavy Metals in Fish Muscles and Health Risk for Humans

The results showed that the fish species did not differ in the concentration of Pb and Cd within the lakes. The Pb content ranged from 0.0305 mg kg^−1^ for *E. lucius* in Lake Syczyńskie to 0.159 mg kg^−1^ for *R. rutilus* from Lake Czarne Sosnowickie. Similarly, in the case of Cd, there were no statistical differences in the content of this metal in fish muscles, and its content ranged on average from 0.0010 mg dm^−1^ for *E. lucius* from lakes Dratów and Syczyńskie to an average of 0.0095 mg dm^−1^ for *R. rutilus* from Lake Czarne Sosnowickie (Table 4). The greatest variation in the Hg content was noted for the studied species from lakes Dratów and Czarne Sosnowickie. In both of these lakes, the highest concentration of Hg was found for *P. fluviatilis*, although in Lake Czarne Sosnowickie it was over twice as high (*p* < 0.05) (Table 4).

The comparison of individual fish species between the lakes showed that Hg concentrations several times higher were recorded for *R. rutilus* and *P. fluviatilis* from Lake Czarne Sosnowickie (*p* < 0.05). Additionally, in the case of Cd, more than four times higher concentrations (average 0.0095 mg kg^−1^) of this heavy metal were recorded for the *R. rutilus* muscles from Lake Czarne Sosnowickie (*p* < 0.05) (Table 4).

An analysis of the content of the tested heavy metals in the fish muscles in terms of food guilds showed that higher values of Pb and Cd were recorded only in non-predatory fish (*R. rutilus*) from Lake Czarne Sosnowickie (*p* < 0.05) (Figure 2). The greatest differentiation was noted for Hg, where the highest values were also found for non-predatory fish, but increased values of this heavy metal were also noted for predatory fish from lakes Czarne Sosnowickie and Syczyńskie (*p* < 0.05) (Figure 2).

The evaluation of the dependence of the content of the tested heavy metals in fish, water, and sediments allowed us to conclude that the obtained correlation indicators reached low values. However, there was a significant negative correlation (correlation coefficient = −0.6856) between the content of Cd in sediments and fish tissues in Lake Dratów. A positive relationship between the Pb content in water and sediments was proved in Lake Syczyński (correlation coefficient = 0.6886, *p* < 0.05). Moreover, the analyses confirmed a significant positive correlation (correlation coefficient = 0.6746) between the content of Cd and Pb in fish muscles of Lake Czarne Sosnowickie.

The content of the heavy metals, i.e., Pb, Cd and Hg in the muscles of the studied fish posed no threat to the health of consumers THQ and TTHQ < 1 (Table 5).

## 4. Discussion

The monitoring of heavy metal contamination is one of the main challenges in ensuring food safety of animal origin. Heavy metals like Pb, Cd, and Hg are toxic elements, characterised by the highest accumulation factor, but the presence of which in food is largely dependent on the condition of the environment [36]. Exposure to the impact of metallic pollutants is especially important in the eating of fish, the integration of which with the living environment and the biomagnification process, may pose a threat to the consumer’s health [37,38]. The issues related to the protection and improvement of the condition of water ecosystems are included in the Directive 2000/60/EC of the European Parliament and of the Council of 23 October 2000 (Journal of Laws UE L of 22 December 2000) [39] known as the Water Framework Directive. According to this regulation, the heavy metals that are the subject of this study (Pb, Cd, and Hg) are priority substances in the field of water policy, which are considered to be particularly harmful to the aquatic environment. The maximum permissible concentration of metals in lake water, which should not be exceeded due to the protection of human health and the environment, is specified in the Regulation of the Minister of Maritime Economy and Inland Navigation of 11 October 2019 (Journal of Laws of 2018, item 2268 and of 2019, items 125, 534 and 1495) [40]. In the present study it was found that the limit value established for Hg was exceeded in all tested lakes (>0.07 µg dm^−3^) and Pb in water samples from lakes Syczyński and Czarny Sosnowickie (>4 µg dm^−3^). The concentration of Cd in all analysed samples was within limits permitted by the regulation (0.45–5 µg dm^−3^); however, it should be noted that in the case of Cd and its compounds, the quality standards depend on water hardness, but the authors did not determine this parameter.

The quality of lake waters in terms of pollution with priority substances is assessed under the national program “Monitoring of surface waters” [41]. According to the data available in the reports for 2014–2020, a small part of the lakes from the Łęczyńsko-Włodawskie Lakeland was monitored. Additionally, heavy metal pollution measurements were made only in some of the examined reservoirs. However, it should be noted that the maximum allowable concentrations of Pb, Cd, and Hg were not exceeded in any of the lakes studied. As shown by the review of the available sources, the content of Pb and Cd in the water samples covered by these studied lakes corresponded with the results obtained by Gwoździński et al. [42], who studied the waters of unmonitored lakes in Bory Tucholskie near the southwestern border of Wdzydzki Landscape Park. On the other hand, the concentrations of heavy metals found in the present study were much lower than those found in the water of lakes Miedwie and Dębie in North-West Poland [43,44].

The analysis of bottom sediments showed a significant quantitative differentiation of the examined metals between individual lakes and their significantly higher content in the sediments compared to the water concentrations. The present study results are consistent with the observations of other authors, according to which, heavy metal pollutants getting into surface waters are deposited mainly in bottom sediments. This mechanism is influenced by the sediment structure, which, unlike the water column, is stable and immobile, preventing rapid resuspension and biochemical dissolution of metallic bonds back in the water column [45,46,47].

Assessment of the degree of contamination of bottom sediments with heavy metals based on the geochemical criterion according to the Bojakowska classification allows the sediments of Lake Dratów to be classified as class I which includes those sediments considered as unpolluted [48,49]. It should be noted that the content of all investigated metals in the bottom sediments of this lake did not exceed the geochemical background determined for Pb, Cd, and Hg, amounting to 15, <0.5, and 0.005 mg kg^−1^, respectively. In the sediments of the lakes Syczyńskie and Czarny Sosnowickie, the geochemical background for the lead was exceeded more than twice and three times, respectively, while the Cd and Hg content were below the limit value. Since the sediment is assessed as polluted even when the limit content is exceeded only for one element, the sediments from these lakes should be classified as class II, moderately polluted, where the harmful impact on living organisms is sporadic. The obtained results correspond to the results of Szafran [50], which compared the content of heavy metals in the bottom sediments of three shallow lakes in Polesie Lubelskie, including Lake Syczyńskie. The author showed that the content of Pb and Cd in the sediments of Lake Syczyńskie in 2003 were respectively: 38.23 and 0.6 mg kg^−1^. Compared to the present study, such convergent results obtained over several years indicate a lack of progressive anthropopressure within the reservoir. Moreover, Tyllman et al. [51], examining the content of heavy metal contents in bottom sediments from twenty-three lakes of North-eastern Poland, showed that lakes from the Masurian and Suwalki Lakelands are characterised by a higher Cd content in the sediments, with much lower contents of Pb.

The content of heavy metals in fish depends on many factors: i.e., fish species, age, nutrition, inhabited habitat, and as well as of the condition of the aquatic environment [10,52,53,54]. Łuczyńska and Brucka-Jastrzębska [55], analysing the content of heavy metals in the muscles of *R. rutilus*, *E. lucius,* and *P. fluviatilis* from four lakes of the Olsztyn Lake District (Poland), stated that the content of Pb in the muscles of these species was higher compared to their contents for the same fish species caught from lakes Dratów and Syczyńskiego and for *P. fluviatilis* from Lake Czarne Sosnowickie. Moreover, fish from the Olsztyn Lake District were also characterised by a higher Cd content and more than 10 times higher Hg content than the present study results. The exceptions were the muscles of *R. rutilus* from Lake Czarne Sosnowickie, where the content of Cd was more than three times higher compared to the highest content of this metal found for this species in Lake Pluszne. Much higher Cd content in the muscles of the studied fish species was also noted in fish caught from Lake Dzierżno Duże Dam Reservoir [56], and the higher Hg amount in the muscle tissue of perch caught in drinking-water reservoirs located in the Morava River Basin was noted by Novotna Kruzikova et al. [57]. The same authors found that the Hg content in fish muscles depends not only on the fish species, but also on where they are caught. The differences in the Hg and Cd content between the places where the fish were obtained were also found in the present study results.

According to the study of Svobodová et al. [58], there is a relationship between the accumulation of Hg in fish tissues and organs and the length of the food chain: predatory fish (*E. lucius*) > benthophagous with a significant proportion of fish in food (*P. fluviatilis*) > typical benthophagous > planktonophagous fish species (*R. rutilus*). Other authors also suggest that fish’s mercury content depends on habitat and eating habits [59,60]. As in the current research, Łuczyńska et al. [61] also showed a higher Hg content in perch tissues than the content of this element in *R. rutilus* tissues. However, it should be noted that significant differences in the Hg content between these fish species were found only in Lake Dratów. Our observations did not fully confirm the relationships mentioned above. For example, *E. lucius* muscles contained slightly higher contents of this element than planktonophagous fish *R. rutilus* caught from lakes Dratów and Syczyńskie, while significantly lower contents for this species were found from Lake Czarne Sosnowickie. Despite the commonly known phenomenon of the biomagnification of heavy metal contaminants of fish, studies by Liu et al. [62] showed that the content of heavy metals in fish tissues depends a lot on the balance between the uptake rate and the elimination rate. Moreover, this depends on the environmental conditions and the life span of the fish. Younger individuals who are characterised by higher metabolic activity and food intake rate may accumulate metallic impurities in their tissues and organs to a greater extent than older fish. In addition, studies by Kh et al. [63] identified several species-specific factors such as feeding behaviours, swimming patterns, genetic tendency, and/or other factors such: as age and geographic distribution. The latter factor influenced the differences in the accumulation of heavy metals in fish, even of the same species.

Fish as food should conform to requirements as defined in the EU food legislation, i.e., acceptable content of Pb, Cd, and Hg [64]. The permitted maximum concentration of both Pb and Cd in the muscles of the three studied fish species is 0.30 and 0.050 mg kg^−1^ by weight, respectively, whereas the maximum concentration of Hg in muscles of *R. rutilus* and *P. fluviatilis* were 0.50 mg kg^−1^ by weight, and in the muscles of *E. lucius,* 1.0 mg kg^−1^ by weight. Pb, Cd, and Hg concentration in the muscles of *R. rutilus*, *E. lucius,* and *P. fluviatilis* from the Dratów, the Syczyńskie, and the Czarne Sosnowickie lakes did not exceed the acceptable values. According to the results of studies by other authors, the content of Pb, Cd, and Hg in the muscles of various food fish species was 0.03–8.62 mg kg^−1^ by weight [65,66], 0.02–0.74 mg kg^−1^ by weight [65,67,68], and 0.056–0.89 mg kg^−1^ by weight [69,70], respectively. The comparison of Pb, Cd, and Hg content in the muscles of the three studied fish species with the content of such metals in muscles of various food fish species indicated that the content of said metals in the muscles of the three studied fish species was low.

The health risk assessment concerning the content of Hg, Pb, and Cd in muscles of the three studied fish species sampled from three lakes was conducted assuming estimated daily intake and reference doses—and calculating THQ and TTHQ [35]. The acceptable value of THQ and TTHQ is 1. In all of the fish studied, THQ and TTHQ values were smaller than 1, which means that the heavy metals of interest, i.e., Pb, Cd, and Hg, posed no threat to the health of consumers ingesting such fish species, neither individually (THQ) nor collectively (TTHQ).

## 5. Conclusions

Analysis of Pb, Cd, and Hg concentrations in water and bottom sediments, despite significant differences found in the sediments of the studied lakes, indicate a slight influence of anthropopressure. The significantly higher concentrations of heavy metals (with the exception of Pb in Lake Dratów) in the bottom sediments indicate that this element of the environment of the studied lakes was the main factor for determining the bioaccumulation of these metals in fish depending on its eating habits. The concentration of Hg, Pb, and Cd in the muscles of the *R. rutilus*, *E. lucius,* and *P. fluviatilis* from the Dratów, Syczyńskie, and Czarne Sosnowickie lakes were below the acceptable levels as defined in the Commission Regulation (EC) 1881/2006. The EDI, THQ, and TTHQ values did not indicate a risk of the toxic effect of Hg, Pb, and Cd on people consuming the meat of analysed fish species.

## Figures and Tables

**Figure 1 animals-11-03507-f001:**
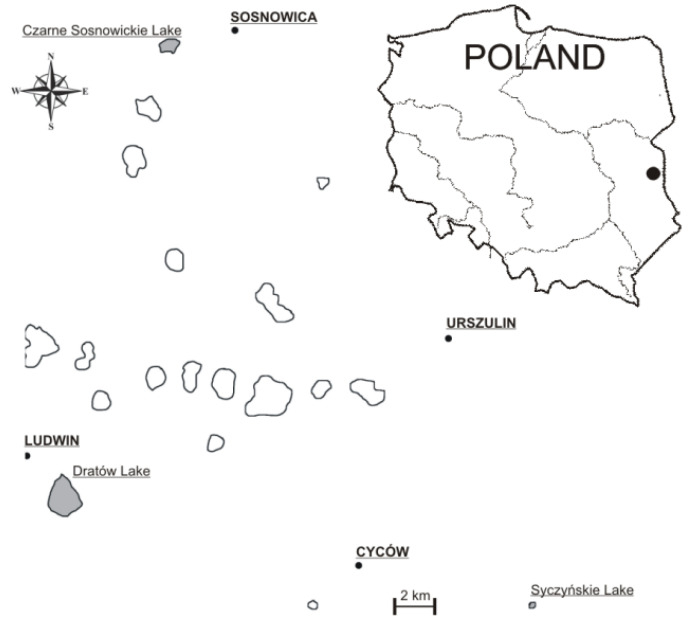
The locations of the studied lakes.

**Figure 2 animals-11-03507-f002:**
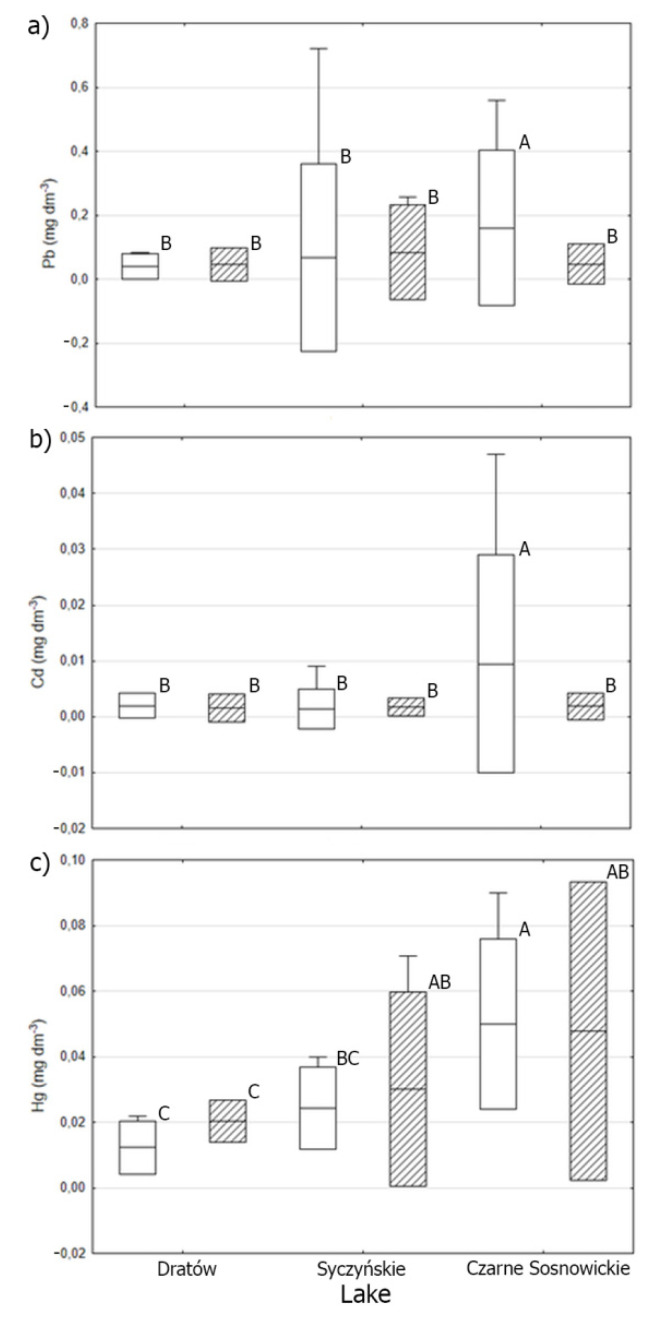
The concentration of Pb (**a**), Cd (**b**), and Hg (**c**) in muscles (in mg kg^−1^) of non-predatory fish (box without pattern) and predatory fish (box with pattern) in studied lakes; explanation of graphics: whiskers—minimal and maximal values, line—mean value, and box—standard deviation. The different letters on the box—differences are statistically significant.

**Table 1 animals-11-03507-t001:** Characteristics of the studied lakes.

Lake	Dratów	Syczyńskie	Czarne Sosnowickie
Parameters	Mean ± SD	Mean ± SD	Mean ± SD
Surface area (ha)	167.9	5.7	39.0
Max. depth (m)	3.3	4.0	15.6
pH *	8.32 ± 0.31	7.94 ± 0.61	7.44 ± 0.25
Oxygen (mg O_2_ dm^−3^) *	10.65 ± 0.56	11.69 ± 3.28	7.20 ± 2.37
N-NH_4_ (mg NH_4_ dm^−3^) *	0.085 ± 0.069	0.548 ± 0.283	0.835 ± 0.095
N-NO_3_ (mg NO_3_ dm^−3^) *	0.096 ± 0.112	0.185 ± 0.238	0.636 ± 0.202
P-PO_4_^3−^ (mg PO_4_^3−^dm^−3^) *	0.024 ± 0.020	0.232 ± 0.185	0.037 ± 0.031
P_tot_ (mg P dm^−3^) *	0.191 ± 0.010	0.499 ± 0.098	0.146 ± 0.057
Chlorophyll a (mg dm^−3^) *	79.06 ± 41.64	73.84 ± 39.24	14.52 ± 10.01
TOC (mg C dm^−3^) *	6.7 ± 0.7	7.1 ± 1.4	24.5 ± 4.9
Trophic status	eutrophic	hypertrophic	eutrophic
Water mixing type	polimyctic	polimyctic	polimyctic
Fishery lake type	tench-pike	tench-pike	tench-pike
Used by	PAA Lublin	PAA Lublin	PAA Chełm

* Mean for 2012–2013 years; SD—standard deviation.

**Table 2 animals-11-03507-t002:** The total length (TL, in mm) and body mass (W, in g) of analysed fish and participation of these species in fishing by anglers (%) in 2014–2018 in studied lakes.

Lake	Fish Species	N	Total Length		Body Mass		Participation in Results of Angling (%)	
			Mean ± SD	Min–Max	Mean ± SD	Min–Max	In Abundance	In Biomass
Dratów	*R. rutilus*	11	16.38 ± 2.41	13.00–20.50	51.30 ± 27.23	18.85–110.68		
	*E. lucius*	4	35.75 ± 0.87	35.00–36.50	361.63 ± 38.83	328.00–395.25	7.39 ± 8.47	7.16 ± 5.43
	*P. fluviatilis*	6	16.83 ± 4.48	12.00–22.00	45.56 ± 40.30	10.11–96.27		
Syczyńskie	*R. rutilus*	22	16.40 ± 1.99	13.80–23.50	53.88 ± 26.89	26.44–160.61		
	*E. lucius*	4	32.25 ± 2.60	30.00–34.50	257.79 ± 83.38	185.58–330.00	79.70 ± 26.25	73.23 ± 14.91
	*P. fluviatilis*	10	13.48 ± 1.66	11.00–17.60	27.39 ± 13.92	12.85–63.87		
Czarne Sosnowickie	*R. rutilus*	24	16.60 ± 2.79	13.00–26.00	47.23 ± 38.08	16.10–195.04		
	*E. lucius*	4	35.50 ± 0.71	35.00–36.00	397.00 ± 4.24	394.00–400.00	22.39 ± 11.35	30.94 ± 9.62
	*P. fluviatilis*	5	23.10 ± 4.98	15.00–28.50	180.57 ± 110.54	41.11–346.58		

N—number of fish, SD—standard deviation.

**Table 3 animals-11-03507-t003:** The data on the angling use of the studied lakes in the years 2014–2018.

Lake	Dratów	Syczyńskie	Czarne Sosnowickie
Parameters	Mean ± SD	Mean ± SD	Mean ± SD
Total number of fish caught by anglers (ind. year^−1^)	3807.80 ± 566.73	245.52 ± 165.94	468.80 ± 279.58
Total biomass of caught fish by anglers (kg year^−1^)	1663.54 ± 832.25	147.80 ± 217.14	189.36 ± 124.23
Participation of predatory fish in abundance (%)	5.76 ± 6.87	48.72 ± 42.75	13.29 ± 4.32
Participation of non-predatory fish in abundance (%)	94.24 ± 6.87	51.28 ± 42.75	86.71 ± 4.32
Participation of predatory fish in biomass (%)	18.12 ± 11.93	47.17 ± 29.44	45.02 ± 6.86
Participation of non-predatory fish in biomass (%)	81.88 ± 11.93	52.83 ± 29.44	54.98 ± 6.86
Number of fish on ha of the lake (ind. year^−1^)	22.68 ± 3.38 ^b^	43.07 ± 29.11 ^a^	12.02 ± 7.17 ^b^
Biomass of fish on ha of the lake (kg year^−1^)	9.91 ± 4.96	25.93 ± 38.10	4.86 ± 3.19
Number of anglers on ha of the lake (pers. year^−1^)	1.37 ± 0.57 ^b^	2.63 ± 0.28 ^a^	0.93 ± 0.19 ^b^
Number of fish per capita (ind. angler^−1^ year^−1^)	19.55 ± 9.87	16.31 ± 10.28	12.37 ± 5.87
Biomass of fish per capita (kg angler^−1^ year^−1^)	7.11 ± 1.19	9.88 ± 14.45	4.92 ± 2.06

Means in a row with the same letter—no statistical differences.

**Table 4 animals-11-03507-t004:** The contents of Pb, Cd, and Hg in water (in mg dm^−3^), sediment and fish muscles (in mg kg^−1^) from studied lakes.

Lake	Fish Species	Dratów	Syczyńskie	Czarne Sosnowickie
Heavy Metal		Mean ± SD	Mean ± SD	Mean ± SD
Pb	*R. rutilus*	0.0399 ± 0.0200	0.0672 ± 0.1472	0.1595 ± 0.1215
	*E. lucius*	0.0360 ± 0.0010	0.0305 ± 0.0064	0.0920 ± 0.0014
	*P. fluviatilis*	0.0533 ± 0.0325	0.1057 ± 0.0787	0.0296 ± 0.0050
	Water	0.0046 ± 0.0007	0.0189 ± 0.0079 ^b^	0.0220 ± 0.0400 ^b^
	Sediment	2.5852 ± 0.9636 ^C^	38.2960 ± 1.2839 ^aB^	45.8234 ± 10.5580 ^aA^
Cd	*R. rutilus*	0.0020 ± 0.0011 ^B^	0.0014 ± 0.0018 ^B^	0.0095 ± 0.0098 ^A^
	*E. lucius*	0.0010 ± 0.0010	0.0010 ± 0.0001	0.0015 ± 0.0007
	*P. fluviatilis*	0.0020 ± 0.0015	0.0020 ± 0.0008	0.0020 ± 0.0014
	Water	0.0012 ± 0.0002 ^b^	0.0020 ± 0.0006 ^b^	0.0030 ± 0.0008 ^b^
	Sediment	0.2940 ± 0.0463 ^aB^	0.4337 ± 0.0878 ^aA^	0.4769 ± 0.1563 ^aA^
Hg	*R. rutilus*	0.0123 ± 0.0040 ^bB^	0.0243 ± 0.0063 ^B^	0.0499 ± 0.0130 ^aA^
	*E. lucius*	0.0185 ± 0.0000 ^b^	0.0255 ± 0.0081	0.0215 ± 0.0007 ^b^
	*P. fluviatilis*	0.0216 ± 0.0037 ^aB^	0.0319 ± 0.0168 ^B^	0.0583 ± 0.0171 ^aA^
	Water	0.0016 ± 0.0004 ^bB^	0.0010 ± 0.0002 ^B^	0.0050 ± 0.0008 ^bA^
	Sediment	0.0062 ± 0.0001 ^aB^	0.0012 ± 0.0002 ^C^	0.0211 ± 0.0001 ^aA^

Lowercase letters in columns—differences within the lake, uppercase letters—differences between lakes, and means marked with different letters differ statistically significant at *p* < 0.05.

**Table 5 animals-11-03507-t005:** The estimated daily intakes (EDI) of the metals by the consumption of the examined fish and hazard quotient (THQ and TTHQ).

Fish Species	Metal	Range of Concentration (µg g^−1^ ww)	RfD (µg kg^−1^ per day)	EDI (µg kg^−1^ per day)	THQ/TTHQ *
*R. rutilus* *E. lucius* *P. fluviatilis*	Pb	0.0399–0.1595 0.0305–0.0920 0.0296–0.1057	4	0.0031–0.0125 0.0024–0.0072 0.0023–0.0083	0.0008–0.0031 0.0006–0.0018 0.0006–0.0021
*R. rutilus* *E. lucius* *P. fluviatilis*	Cd	0.0014–0.0095 0.0010–0.0015 0.0020	1	0.0001–0.0007 0.0001–0.0001 0.0002	0.0001–0.0007 0.0001–0.0001 0.0002
*R. rutilus* *E. lucius* *P. fluviatilis*	Hg	0.0123–0.0499 0.0185–0.0255 0.0216–0.0583	0.1	0.0010–0.0039 0.0014–0.0020 0.0017–0.0046	0.0100–0.0390 0.0140–0.0200 0.0170–0.0460

* TTHQ for total Pb, Cd, and Hg in the muscles of *R. rutilus, E. lucius and P. fluviatilis* ˂ 1.

## Data Availability

Data on the results of the analysis of heavy metals are available on request from the authors of article. The data on the fishing use of the lakes are available from the lakes users-PPA Lublin and PPA Chełm.

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
