# Peer review of "A Food-Safety Risk Assessment of Mercury, Lead and Cadmium in Fish Recreationally Caught from Three Lakes in Poland"

_animals, 2021, doi:10.3390/ani11123507_

Round 1
Reviewer 1 Report
Dear autors,
it was interesting to read your MS. Generally, the paper is well wrtitten, referring to earlier results.
The aims are good defined. The Resukts chapter is well structured and clear. Figures and Tables follow the text in a good way. Tables are god structured, informative and clear. Statistical analyses are adequate.
My suggestion is, to improve MS, use some other reference, more appropriate. All comments and corrections are given in the text, as track-changes.
The chapter Disscussion can be extended with an emphasis on impact of heavy metal content on anglers health, only sentences from 391 to 397 in MS are about it.
And, why As wasn`t include in this study, as toxic one?
In the chapter Introduction, nowhere is mentioned why these three lakes are chosen, which are pollution sources.
Regards.

Author Response
Answers to Reviewers 1 comments
We would like to thank the Reviewer 1 for valuable comments to our manuscript. Below are the responses to the Reviewer's comments.
Whole manuscript has been verified by a native speaker. Any changes are marked in the text in track changes modecomment
“The chapter Disscussion can be extended with an emphasis on impact of heavy metal content on anglers health, only sentences from 391 to 397 in MS are about it.”
The choice of the presented information for this fragment of the discussion section was not coincidental. The one of the aims of the study was to assessment selected species of fish in terms of food safety and it has been presented. We would like to explain that the presented fish species did not pose a health risk, therefore the suggested issue is not related to our results. In our view, the discussion in this respect is pertinent and we would like to leave it as it is.
Comment
„And, why As wasn`t include in this study, as toxic one?”
We agree with the Reviewer that As belongs to absolutely toxic elements, but it was not the aim of our research. In our research, we focused on the toxic elements that are most often studied in freshwater ecosystems, i.e. Pb, Cd and Hg, which allowed for discussion with other authors in this field. We consider the author's suggestion valuable and we will include it in future research.
Comment
„In the chapter Introduction, nowhere is mentioned why these three lakes are chosen, which are pollution sources.”
We decided to add one sentence on lake selection to the chapter Material and Methods sub-section Study area.
„The selected lakes are distant from each other, they have different fishing pressure and are characterized by a different ecological status”
Comment to Simple summary
The whole part has been rewritten and verified by a native speaker
Line 31
„selected and toxic” to sentence was added
Line 35-36
The sentence has been changed and verified by a native speaker
Line 46
„aleo” was replaced by „also”
Line 46 comment:
„in Europe many researcher work on ecotoxicology in freswater, and in fish add other reference”
We agree with Reviewerm that there are many such publications, but the literature used in the work is still very rich (71 items in total), so we did not want to increase the number of sources of literature.
Line 57
The sentence was verified by a native speaker - an "a" was added before the word "human
Line 98
The full Latin name has been added
Line 183
The word „remaining” was replaced by „two other”
Line 223
The typographical error in the word "concentration" has been corrected
Line 244
The species name was added in parentheses
Line 305
Changed as noted by Reviewer
Line 367 and line 373
We would like to note that the reviewer's comment "for this two statements use more proper, better reference" is not clear to us. Due to the fact that the reviewer did not indicate to what extent we should make changes, and what criteria to use to define more relevant publications, we would like to leave them unchanged. At the same time, we would like to emphasize that the studies by Liu et al. and Kh et al. are most relevant to the issues discussed in this part of the discussion.
Line 467
Missing journal name was added
Reviewer 2 Report
In this manuscript, several concerns need to be addressed, more organization and extensive revisions are required as follows:
General comments:
1. There is a problem with using abbreviations throughout the manuscript. The full term should be mentioned first with the abbreviation between paresis then the abbreviations should be used throughout the manuscript. E.g. in line 58 cadmium has been abbreviated as Cd then the full term has been repeated many times throughout the manuscript in lines 82, 272, 292,322, 346, and 399. The same error has been repeated many times for other abbreviations throughout the manuscript.
2. The manuscript is full of grammatical, typographic (E.g. line 82: cadium), formatting, and styling errors. Revision of the manuscript by a native English speaker is highly recommended.
Specific comments:
1. The title needs to be representative of the study as follows:
- The location of the study should be mentioned as it is considered a screening survey.
- The type of fish should be specified or at least define its type freshwater or marine fish.
- It is not recommended to use abbreviations in the title. Use the full term of metals.
2. Abstract need to be rewritten as it is very confusing and not representative of the study in the present form as follow:
- The background should be concise and followed by a clear description of the objective of the study.
- The methods should be briefly described with specific items not general terms like three lakes (line 31), abiotic elements (line 33),…etc. The authors should mention the name of lakes and the type of elements.
- Mention the main results without discussing them and give a clear conclusion of the findings of the study.
3. Introduction: the authors should clarify the novel aspects of the study rather than give general information on the importance of fish and heavy metals hazards.
4. Material and methods:
- Add a map for the locations of the three lakes.
- Table 1: the full term of all abbreviations used within the table should be clarified in the footnote.
- Line 109: describe in detail what superficial water samples and spots of the surface layer of bottom sediments mean? On what depth they have collected.
- Change the subheading " Materials sampling and analysis" to " collection of water, sediment, and fish samples"
- What is the number of collected samples?
- Add another heading of "metal analysis" for lines 133-151".
- Add the references for methods used for metal analysis.
- The quality control should be described in more detail.
- The details of all apparatus and chemicals used should be given as manufacturer, city, and country.
5. Results section is highly recommended to be divided with subheadings.
6. The letter of significance in all tables should be non-capitalized and superscripted. In addition, the indications of the letters should be illustrated in the footnote.
Author Response
Answers to Reviewers 2 comments
We would like to thank the Reviewer 2 for valuable comments to our manuscript. Below are the responses to the Reviewer's comments.
General comments:
Comment 1. There is a problem with using abbreviations throughout the manuscript. The full term should be mentioned first with the abbreviation between paresis then the abbreviations should be used throughout the manuscript. E.g. in line 58 cadmium has been abbreviated as Cd then the full term has been repeated many times throughout the manuscript in lines 82, 272, 292,322, 346, and 399. The same error has been repeated many times for other abbreviations throughout the manuscript.
Throughout the whole text, we have tried to correct the metal names according to a Reviewer 2 comment
Comment 2. The manuscript is full of grammatical, typographic (E.g. line 82: cadium), formatting, and styling errors. Revision of the manuscript by a native English speaker is highly recommended
Whole manuscript has been verified by a native speaker. Any changes are marked in the text in track changes mode
Specific comments:
- The title needs to be representative of the study as follows:
- The location of the study should be mentioned as it is considered a screening survey.
- The type of fish should be specified or at least define its type freshwater or marine fish.
- It is not recommended to use abbreviations in the title. Use the full term of metals.
We believe that the location of the research area should be included in the Material and methods chapter. The subsection Study Area the location of the lakes along with the map were described. Providing the location in the title of MS will unnecessarily extend it, and may additionally be the cause of a comment about the local nature of the research. We added the phrase "from three lakes" to the title, which explains the origin of the fish (freshwater fish) and in some way defines the area of research. In addition, we have given the full names of the metals in the title of the paper.
- Abstract need to be rewritten as it is very confusing and not representative of the study in the present form as follow:
- The background should be concise and followed by a clear description of the objective of the study.
- The methods should be briefly described with specific items not general terms like three lakes (line 31), abiotic elements (line 33),…etc. The authors should mention the name of lakes and the type of elements.
- Mention the main results without discussing them and give a clear conclusion of the findings of the study.
We tried to improve the abstract according to comments from Reviewer 2 and a native speaker.
- Introduction: the authors should clarify the novel aspects of the study rather than give general information on the importance of fish and heavy metals hazards.
In our opinion the Introduction chapter layout is correct. The novel aspects of study was written in the last paragraph (73-79 line) prior to the purpose of the study. In our opinion, the problem should be described more broadly, taking into account the research topics in this area. If we narrowed it down, we could hear a comments from the Reviewers that we focused only on our research and we do not write anything about a very large achievements in the field of heavy metals in the aquatic environment.
- Material and methods - comments:
- Add a map for the locations of the three lakes.
The map (Figure 1) has been added
- Table 1: the full term of all abbreviations used within the table should be clarified in the footnote.
We believe that all unusual abbreviations have been explained, and in our opinion explaining abbreviations of chemical compounds commonly known in hydrobiology and used without explanation in many papers (e.g., N-NH4) is redundant.
- Line 109: describe in detail what superficial water samples and spots of the surface layer of bottom sediments mean? On what depth they have collected.
The depth of water and sediment samples, as suggested by the Reviewer, was determined.
- Change the subheading " Materials sampling and analysis" to " collection of water, sediment, and fish samples
Subheading was changed
- What is the number of collected samples?
In the Materials and Methods chapter the number of water and sediment samples taken for research were determined.
- Add another heading of "metal analysis" for lines 133-151".
New heading was added
- Add the references for methods used for metal analysis.
The new references were added.
- The quality control should be described in more detail.
Quality control has been extended by quantification limit, trueness and repeatability / precision for each element.
- The details of all apparatus and chemicals used should be given as manufacturer, city, and country.
The details of all apparatus and chemicals used in research were added.
- Results section is highly recommended to be divided with subheadings
A subheadings has been added to the Results chapter
- The letter of significance in all tables should be non-capitalized and superscripted. In addition, the indications of the letters should be illustrated in the footnote.
The letters describing significance of the difference has been reduced to superscript format in all tables. In Table 3, we also changed the uppercase letters to lowercase. In Table 4, we used uppercase and lowercase letters because they denote different dependencies, i.e., lowercase letters in columns - differences within the lake, uppercase letters - differences between lakes. Explanations for this are at the end of the table title (see lines 234-235). We believe it is legible.
Reviewer 3 Report
The study did not address the title of the paper. What was the relationship between the water and sediment levels and those in the fish ? Was there any bioaccumulation ? The results on the recreational fish consumption did not have the survey method described ?

Author Response
Answers to Reviewers 3 comments
Thank you for the Reviewer 3 opinion. We will try to respond to comments and doubts of Reviewer 3.
With all due respect, we believe that we have referred to the title of the paper in the research results. In our opinion the Reviewer's remark is unfounded. The Reviewer 3 did not justify in any way why he thought so. We believe that when examining the level of heavy metals in fish tissues, it is good to know the level of these metals in the environment, i.e. in water and bottom sediments that are their immediate surroundings. With our research, we tried to show the dependence of the content of selected heavy metals in water and sediments as well as in fish meat, which can potentially be food for anglers. The latter issue is particularly important from the point of view of food safety, and our research has confirmed the low level of exposure to heavy metals from this food source for humans.
Comment: „The results on the recreational fish consumption did not have the survey method described ?”
We would like to explain that the information has been placed in the manuscript including the reference. Materials and Methods: the subsection 2.3. Health risk assessment (lines 152-167).
The other comments from Reviewer 3 included in the manuscript file
Comment:
„Need to improve English grammar and check for spelling errors through out the paper”.
Whole manuscript has been verified by a native speaker. Any changes are marked in the text in track changes mode.
Comment:
„Do you think that the sediment metals (Pb, Hg and Cd) influenced the muscle levels in the fish ?”
We have tried to check this relationship with our research, but it can be assumed that the living environment (e.g. water and bottom sediments) may affect the level of heavy metals in fish. This is particularly likely because the water is in direct contact with the fish organism and the bottom sediments are the habitat of fish food, especially non-predatory fish (benthivorous fish).
Comment
Is this from a survey of fishermen ? (Table 3)
The data came from the fishing reports of anglers made available to the authors of the study by lake users. This data is collected annually from all anglers fishing in the lakes operated by the PPA. We wrote about the origins of these data in the Material and methods chapter (lines 115-123).
Comment
„Is there any bioaccumulation of the metals ?”
Bioaccumulation of heavy metals in fish tissues depends on many factors, including accumulation from the surrounding environment (water, sediment) and the absorption of metals with food. Our research showed that both the water and sediments of the tested lakes show little contamination with heavy metals. The muscles of the tested fish also contain low levels of lead, cadmium and mercury. This was probably the reason why we were unable to clearly demonstrate the relationship between the content of heavy metals in the muscles of fish and their living environment. In addition, we believe that the low concentrations of heavy metals in the muscles of the studied fish species were also impossible to clearly confirm the relationship between the accumulation of metals in fish tissues and eating habits, as observed by other authors. In the authors' own research, significant differences in the mercury content between fish species belonging to different food guilds were found only in Lake Dratów. Moreover, E. lucius muscles contained a slightly higher content of this element compared to R. rutilus caught from Dratów and Syczyńskie lakes, but much lower than that of this species from Lake Czarne Sosnowickie, which was described by us in the discussion section.
According to the study of the authors cited in the work bioaccumulation in different species of fish occurs with a specific intensity and also changes during the life of the fish and depends on size and age. However, these factors were not taken into account in our research.
Comment
„Why is there a large variation in the concentration of mercury in the fish in lake Czarne Sosnowickie ?”
The answer to this question is difficult. The analysis of the data obtained after assessing the content of heavy metals in the muscles of the fish showed a large variation in the case of Hg in the Lake Czarne Sosnowickie. It seems to us that this could be due to various reasons beyond our control as authors. It may depend on the ecological status and trophic status of the lake, the habitat diversity, the species composition of the fish, and may also depend on the randomly selected fish for analysis. We do not have a definite answer to this question. Please note, that in the case of Cd, the content of this metal in the muscles of the fish is also much greater (min and max) in the case of non-predatory fish (see Figure 2).
Comment
„How come the average concentration is a range of values ? (table 5)”
The mistake has been corrected (range instead of average concentration).
Round 2
Reviewer 2 Report
The manuscript has been improved and no further comments to address.
Author Response
Thanks again for your comments on our manuscript.
At the first reviews the manuscript were already proofreading by a native speaker. We decided that another verification was unfounded and in a so short time. But to cover letter we enclose the statement of translation office with information about proofreading and about the native speaker.
Best regards
Authors

Reviewer 3 Report
Thank you for the clarifications and revisions. A few more edits for your consideration are included. The main one is a revision of the title to better capture the intent of the work and the results.
Example : A food-safety risk assessment of mercury, lead and cadmium in fish recreationally caught from three lakes in Poland.

Author Response
Answers to Reviewer 3 comments after second review
Thank you again for your comments on our manuscript. All comments in the pdf file have been taken into account. According to the comment from Reviewer 3, we also changed the title of the work. We believe that the new title is appropriate and better suited to the topic of the thesis.
At the first reviews the manuscript were already proofreading by a native speaker. We decided that another verification was unfounded and in a so short time. But to cover letter we enclose the statement of translation office with information about proofreading and about the native speaker.
Best regards
Authors
